# Correlation Between Intravascular Platelet Aggregation in Tumors and Hypoxia-Inducible Factor 1 Alpha Expression in Epithelial Ovarian Cancer: Implications for Prognosis and Staging

**DOI:** 10.3390/cancers17030345

**Published:** 2025-01-21

**Authors:** Jung Min Ryu, Yoon Young Jeong, Sun-Jae Lee, Youn Seok Choi

**Affiliations:** 1Department of Obstetrics and Gynecology, School of Medicine, Daegu Catholic University, Daegu 42472, Republic of Korea; medgirl1231@cu.ac.kr (J.M.R.); nning@cu.ac.kr (Y.Y.J.); 2Department of Pathology, School of Medicine, Daegu Catholic University, Daegu 42472, Republic of Korea

**Keywords:** ovarian carcinoma, platelet aggregation, prognosis, tumor hypoxia

## Abstract

This study investigates the association between intravascular platelet aggregation in tumors and the prognosis of patients with ovarian cancer, which is often diagnosed at an advanced stage. The findings suggest that intravascular platelet aggregation in tumors contributes to the formation of a hypoxic environment, which in turn promotes cancer progression, metastasis, and resistance to treatment, leading to poorer survival outcomes. This research has the potential to influence future approaches to managing ovarian carcinoma and improving patient outcomes.

## 1. Introduction

Ovarian carcinoma has the highest mortality rate among gynecological cancers because it is usually diagnosed at an advanced stage. Treatment for ovarian carcinoma typically involves optimal debulking surgery and adjuvant therapy, with the approach determined by the cancer stage, tissue differentiation, the amount of residual tumor, and the presence of alleles associated with homologous recombination deficiency, including BRCA [1,2]. 

Although thrombocytosis is not specific to malignancy, the association between cancer and platelets has been assessed in various studies [3]. Many studies have reported that platelet aggregation occurs more frequently in association with cancer, and the incidence of thrombotic events increases with various cancers, including ovarian cancer [4]. In patients with ovarian carcinoma, especially clear cell carcinoma, the risk of pulmonary thromboembolism or venous thromboembolism is increased [5].

It is known that cancer cells can induce platelet aggregation, which is caused by crucial molecules or surface receptor molecules on the surface of cancer cells [6]. According to a previous study, platelet aggregation in tumor cells is associated with cancer progression and metastasis, leading to research on new platelet-targeted therapeutic strategies [7]. Based on this background, we hypothesized that intravascular platelet aggregation in tumors may induce occlusion of tumor microvessels and hypoxia of the cancer microenvironment in patients with ovarian carcinoma. As a result, cancer cells are subjected to hypoxic stress, which is associated with proliferation, metastasis, and chemoresistance, which may be associated with a poor prognosis. 

However, no association between intravascular platelet aggregation in tumors and prognosis in patients with ovarian carcinoma has yet been reported, and the underlying mechanisms have not been thoroughly elucidated. Therefore, the purpose of this study was to investigate the association between intravascular platelet aggregation in tumors and the prognosis of patients with ovarian carcinoma, along with the underlying mechanisms of this relationship.

## 2. Materials and Methods

### 2.1. Enrolled Patients

We identified patients with ovarian carcinoma who underwent staging surgery, with or without adjuvant chemotherapy, from the gynecologic oncology database managed by the Department of Obstetrics and Gynecology at Daegu Catholic University Hospital. A retrospective review of medical records was conducted for each patient. The inclusion criteria for the patients were as follows: (1) patients with epithelial ovarian cancer, including serous, mucinous, clear cell, and endometrioid histopathological types; (2) patients who received primary treatment at the institution; and (3) patients diagnosed between March 2000 and October 2010. Patients with transitional cell carcinoma, malignant Brenner tumors, or carcinosarcomas were excluded. Additionally, patients treated with neoadjuvant chemotherapy before primary debulking surgery were also excluded. This retrospective study was approved by the institutional review board (IRB) of our medical center (IRB approval number: CR-21-089, 16 June 2021). Based on the above criteria, a total of 144 patients with ovarian carcinoma were enrolled in the study. 

### 2.2. Construction of Tissue Microarrays (TMA)

Before the samples were prepared for tissue microarrays (TMA), representative paraffin tumor blocks were selected based on the initial evaluation of hematoxylin and eosin (H&E)-stained slides of ovarian carcinoma tissues. Two tumor tissue cores (2 mm in diameter) were extracted from each donor ovarian carcinoma block using a manual punch arrayer (Quick-Ray, Uni-Tech Science, Seoul, Republic of Korea). These cores were placed into a new recipient paraffin block, which ultimately contained 59–91 tissue cores. Multiple sections (5 µm thick) were cut from the TMA blocks and mounted onto microscope slides. The TMA H&E-stained sections were examined under a light microscope by a pathologist to verify the presence of representative tumor areas.

### 2.3. Immunohistochemical Staining (IHC)

Immunohistochemistry was performed on 5 µm thick TMA tissue sections using the Ventana BenchMark ULTRA (Ventana Medical Systems Inc., Tucson, AZ, USA) according to the manufacturer’s instructions. CD42b, also known as platelet glycoprotein1b alpha chain (GP1BA), is expressed on platelets and megakaryocytes [8]. Thus, an anti-CD42b antibody was used for IHC staining to identify platelet aggregation in ovarian carcinoma. The specimens were incubated with a primary rabbit anti-CD42b antibody (1:100; ab134087; Abcam, Cambridge, MA, USA). Subsequently, the sections were incubated with the following primary antibodies: anti-hypoxia-inducible factor-1α (HIF-1α, mouse monoclonal, 1:100, ab16066, Abcam, Cambridge, MA, USA), anti-vascular endothelial growth factor A (VEGF, mouse monoclonal, 1:100, ab1316, Abcam, Cambridge, MA, USA), and anti-platelet-derived growth factor B (PDGF, mouse monoclonal, 1:100, ab51869, Abcam, Cambridge, MA, USA).

### 2.4. Interpretation of the IHC

The stained slides were scanned using a virtual slide scanner (Aperio ScanScope slide scanner, Leica Biosystems, Nussloch, Germany), and the degree of staining of each tissue core was quantified. The IHC result was interpretated as positive for each antibody when the sample was stained brown. CD42b expression levels were graded by an expert pathologist on a scale of 1 to 3 based on intravascular platelet aggregation in ovarian carcinoma. The patients’ medical records were blinded during the assessment. The CD42b staining was graded as follows: grade 1, no intravascular platelet aggregation; grade 2, intravascular platelet aggregation in tumors; and grade 3, intravascular microthrombus in tumors.

HIF-1α, PDGF, and VEGF expression levels were graded on a scale of 0 to 3 based on cytoplasmic and membrane staining intensity and the proportion of positive tumor cells by an expert pathologist (one of our authors, Sun Jae Lee). The staining was graded as 0 if no malignant cells were stained, 1 if staining was weakly positive in <1/3 of malignant cells, 2 if staining was weakly positive in >2/3 of malignant cells or strongly positive in >1/3 of malignant cells, and 3 if staining was weakly positive in most malignant cells or strongly positive in >2/3 of malignant cells [9].

### 2.5. Definitions of Overall Survival

Overall survival (OS) was defined as the duration from the date of diagnosis to the date of death caused by ovarian carcinoma. Patients who were alive at the time of analysis were censored at their most recent follow-up.

### 2.6. Statistical Analysis

Variables between groups were compared using the chi-square test. Correlation analysis of categorical data was performed using Spearman’s correlation analysis. The OS was estimated using the Kaplan−Meier life-table method. Differences in survival rates were evaluated using log-rank tests. The *p*-values were obtained from two-sided tests, and statistical significance was set at *p* < 0.05. Statistical analysis was performed using SPSS ver. 21 (IBM Co., Armonk, NY, USA).

## 3. Results

The characteristics of the enrolled patients are summarized in Table 1. These include age, parity, menopause, preoperative platelet count, CA125, CA19-9, largest ovarian tumor diameter, tumor site, stage, pathologic type, presence of hypertension, diabetes mellitus, ascites, and pelvic and para-aortic lymph node metastasis. A total of 144 patients with ovarian carcinoma were included in this study. The number of patients in each ovarian carcinoma stage was as follows: stage I—51, stage II—35, stage III—68, and stage IV—11. Regarding histologic type, serous adenocarcinoma was the most common (43.8%), followed by mucinous adenocarcinoma (24.3%), endometrioid adenocarcinoma (19.4%), and clear cell carcinoma (12.5%). 

The IHC staining grades for intravascular platelet aggregation in ovarian carcinoma were categorized into three grades: grade 1 staining was observed in 17.4% (n = 25) of patients, grade 2 in 59.0% (n = 85), and grade 3 in 23.6% (n = 23.6). The higher grades of IHC staining corresponded to greater degrees of intravascular platelet aggregation. Representative pathological photographs are presented in Figure 1. 

The association between intravascular platelet aggregation grade and the clinicopathological characteristics of ovarian carcinoma is presented in Table 2. There was no statistically significant association between intravascular platelet aggregation grade and presence of bilaterality, ascites, involvement of pelvic and para-aortic lymph nodes, or pathologic type of ovarian carcinoma. Blood thrombocytosis and platinum chemotherapy responses were also not associated with platelet aggregation grade. However, in patients with more advanced stages of ovarian cancer, higher levels of platelet aggregation were observed in the blood vessels within the ovarian carcinoma tissue (*p* = 0.002, Spearman’s correlation analysis). 

Table 3 presents the results of the Kaplan−Meier survival analysis conducted based on CD42b IHC staining grades. Univariate analysis was performed to evaluate the association between the clinicopathologic characteristics of patients with ovarian carcinoma and survival rates. The results showed that age, ascites, pelvic or para-aortic lymph node metastasis, stage, and grade of intravascular platelet aggregation of the tumor were statistically significantly related to prognosis. The grade of intravascular platelet aggregation in ovarian carcinoma was statistically significantly associated with a poor prognosis (*p* = 0.037).

Based on the results shown in Table 3, multivariate analysis was performed for age (≤50 vs. >50 years), the presence of ascites and pelvic or para-aortic lymph node metastasis, stage (I or II vs. III or IV), and intravascular platelet aggregation in tumors (grade 1 vs. grades 2 and 3) by using a Cox regression hazard model. Age and stage were identified as independent prognostic factors for the survival of patients with ovarian carcinoma, as follows: age ≤50 vs. >50 (adjusted HR 1.940, 95% CI 1.197–3.146, *p* = 0.007) and stage I or II vs. III or IV (adjusted HR 6.874, 95% CI 3.369–14.025, *p* < 0.001). However, the presence of ascites, pelvic or para-aortic lymph node metastasis, and intravascular platelet aggregation in tumors were not independent prognostic factors. This result was found even when intravascular platelet aggregation in tumors was analyzed as grade 1 or 2 vs. grade 3.

The results presented in Table 4 indicate that additional analyses were performed to investigate the mechanism underlying the association between intravascular platelet aggregation in tumors and survival. These analyses revealed a significant correlation between intravascular platelet aggregation in tumors and HIF-1α expression (correlation coefficient = 0.226, *p* = 0.006). PDGF and VEGF expression were also analyzed, but no significant correlations were observed for PDGF (correlation coefficient = 0.034, *p* = 0.686) or VEGF (correlation coefficient = 0.131, *p* = 0.116). Figure 2 presents representative results of IHC staining for HIF-1α, VEGF, and PDGF, illustrating the staining grades 0 to 3, which reflect varying expression levels in the tumor samples.

## 4. Discussion

The association between cancer and platelets has been consistently reported [3]. Thrombocytosis often arises as a reactive condition triggered by factors such as acute infection, tissue damage, chronic inflammation, surgery, iron deficiency, or the rebound effect after bone-marrow suppression. Although thrombocytosis is not specific to cancer, the association between platelets and cancer has been increasingly recognized. In patients without anemia or inflammation, thrombocytosis may indicate the presence of occult malignancies [10].

Cancer enhances thrombopoiesis in the liver, leading to increased platelet production in the bone marrow. The production of hepatic thrombopoietin and subsequent thrombocytosis is caused by elevated levels of thrombopoietin-inducing cytokines, such as interleukin-1 (IL-1), IL-3, IL-11, and tumor-derived IL-6, in tumor-host tissues [11]. In a mouse model, hepatic thrombopoietin synthesis is upregulated in response to tumor-derived IL-6 by the mechanism of paraneoplastic thrombocytosis [11]. Additionally, granulocyte-macrophage colony-stimulating factor (GM-CSF) and granulocyte colony-stimulating factor (G-CSF) also promote the production of platelets [12,13].

According to the results in Table 3, intravascular platelet aggregation in tumors is associated with poor prognosis in patients with ovarian carcinoma. Additionally, based on the results in Table 2, intravascular platelet aggregation in tumors was found to be associated with advanced stage in patients with ovarian carcinoma. This suggests that poor prognosis may also be associated with an advanced stage of ovarian carcinoma. However, according to the results of the multivariate analysis, intravascular platelet aggregation does not appear to be a direct survival factor in ovarian carcinoma. Based on these results, it is suggested that as the stage progresses, platelet aggregation within tumor blood vessels and subsequent exposure to hypoxia may impact prognosis. There are studies on the association between tumor platelet infiltration and the prognosis of patients with pancreatic and gastric cancers [14,15,16]. One study has shown that platelet aggregation induced by tumor cells protects tumor cells from tumor necrosis factor-α mediated cytotoxicity, which helps tumor cells evade the body’s immune system [17]. However, difference in prognosis may also possibly depend on the proangiogenic role of platelets in the tumor microenvironment [18].

Several studies have reported that thrombocytosis is associated with poor prognosis in various malignancies. However, some studies have found conflicting results, so the association between thrombocytosis and prognosis is currently uncertain [19,20,21,22,23]. According to previous studies, ovarian carcinoma and thrombocytosis are associated with more advanced forms of the disease and lower survival rates. It is known that in patients with various malignancies, the probability of thrombotic event is increased [24]. Approximately 10% to 15% of patients with malignancies developed cancer-related venous thromboembolism, deep vein thrombosis, or pulmonary thromboembolism [25,26]. This increased risk is attributed to hypercoagulability caused by platelet aggregation [4,6,27]. 

Cancer changes platelet behavior by triggering platelet granule release and modifying the platelet phenotype and RNA profile. In particular, cancer cells induce platelet aggregation, and this ability is correlated with increased potential for cancer-cell metastasis [6,28]. We hypothesized that platelet aggregation or microthrombus formation within blood vessels contributes to a hypoxic tumor microenvironment. Furthermore, hypoxic stress in cancer cells induced by platelet aggregation or microthrombus formation in intertumoral blood vessels may be associated with the prognosis of patients with ovarian carcinoma. A previous study reported that hypoxic stress leads to the secretion of hypoxia-inducible factors, including platelet-derived growth factor B, angiopoietin 2, angiopoietin-like 4, placental growth factor, and vascular endothelial growth factor [29]. These factors promote cancer-cell proliferation, metastasis, and chemoresistance, which can lead to a poor prognosis for patients. However, the mechanism underlying the association between intravascular platelet aggregation in tumors in ovarian carcinoma and poor prognosis is still unclear.

Based on the correlation between HIF-1α and intravascular platelet aggregation in tumors in this study, it can be suggested that intravascular platelet aggregation within tumors induces changes in the tumor microenvironment, promoting hypoxic damage within the tumor that may contribute to reduced survival rates in cancer patients [30,31]. Intravascular platelet aggregation within tumors has the potential to activate HIF-1α, which contributes to enhancing drug resistance by promoting cellular metabolic changes and inhibiting cell death [32]. Several previous studies have found that hypoxic stress is associated with chemoresistance and metastasis in cancer cells [33,34,35]. In other studies, HIF-1α inhibitors have also been reported as therapeutic targets for inhibiting tumor proliferation and malignancy [36,37]. Therefore, additional therapeutic approaches targeting HIF-1α or platelet aggregation may have the potential to improve the prognosis of patients with ovarian carcinoma.

This study has several limitations. First, as a retrospective study relying on previously collected data, it may lack consistency or completeness. Second, the patients included in this study were treated between 2000 and 2010, a relatively long time ago, which may reduce the applicability of our findings to the current treatment landscape. While updated data from more recently treated patients could provide additional insights, the shorter follow-up periods associated with these patients may make it hard to perform a comprehensive survival analysis. Furthermore, we tried to minimize confounding factors introduced by relatively recent treatment modalities, such as poly (ADP-ribose) polymerase inhibitors, which have significantly influenced patient outcomes. Finally, although our study results show a significant relationship between intravascular platelet aggregation and HIF-1α, the correlation is very weak. This suggests that in addition to the effects of hypoxic damage, other mechanisms may also have had an effect. Therefore, further studies are needed to understand the mechanisms underlying this study result.

## 5. Conclusions

In conclusion, intravascular platelet aggregation in tumors is associated with poor prognosis in patients with ovarian carcinoma. Based on the findings of this study, it can be suggested that hypoxic damage may induce intravascular platelet aggregation in tumors, which is associated with lower survival rates in patients with ovarian carcinoma. Further research will be needed to better understand the underlying mechanisms.

## Figures and Tables

**Figure 1 cancers-17-00345-f001:**
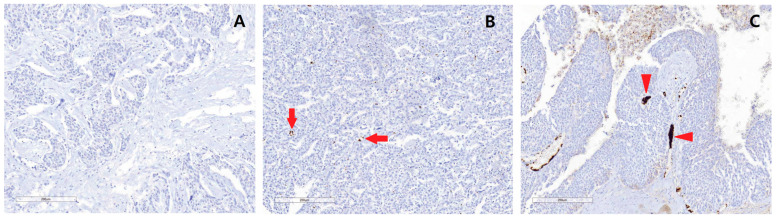
Representative photos of intravascular platelet aggregation in tumors for each grade of CD42b immunohistochemical staining (×100). The staining was graded as follows: grade 1 (**A**), no intravascular platelet aggregation; grade 2 ((**B**), arrows), intravascular platelet aggregation in tumors; grade 3 ((**C**), arrowheads), intravascular microthrombus in tumors.

**Figure 2 cancers-17-00345-f002:**
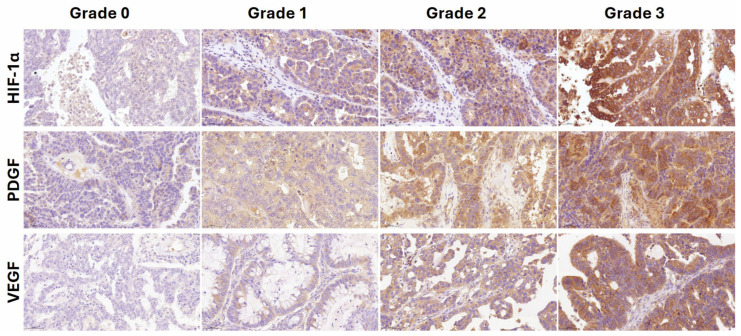
Representative photos of each grade of hypoxia-inducible factor (HIF-1α), platelet-derived growth factor (PDGF), and vascular endothelial growth factor (VEGF) immunohistochemical staining (×200). The staining was graded as 0 if no malignant cells were stained, 1 if staining was weakly positive in <1/3 of malignant cells, 2 if staining was weakly positive in >2/3 of malignant cells or strongly positive in >1/3 of malignant cells, and 3 if staining was weakly positive in most malignant cells or strongly positive in >2/3 of malignant cells.

**Table 1 cancers-17-00345-t001:** Characteristics of the enrolled patients with ovarian carcinoma in this study (n = 144).

Characteristic	Value *
Age (year)	50.1 ± 13.4 (15–78)
Parity	
Nulliparous	18 (12.5%)
Primiparous	15 (10.4%)
Multiparous	111 (77.1%)
Menopause	
Positive	66 (45.8%)
HTN	
Positive	16 (11.1%)
DM	
Positive	6 (4.2%)
Preoperative platelet (10^3^/μL)	292.9 ± 93.7 (50–557)
Preoperative CA125 (U/mL)	733.6 ± 1197.3 (7.5–5000)
Preoperative CA19-9 (U/mL)	191.2 ± 698.0 (1–5000)
Largest ovarian tumor diameter (cm)	11.7 ± 6.9 (2–40)
Tumor site	
Unilateral	71 (49.3%)
Bilateral	73 (50.7%)
Ascites	
Absent	33 (22.9%)
Present	111 (77.1%)
Pelvic LN metastasis	
Negative	105 (72.9%)
Positive	39 (27.1%)
Paraaortic LN metastasis	
Negative	99 (68.8%)
Positive	45 (31.3%)
Stage	
I	51 (35.4%)
II	14 (9.7%)
III	68 (47.2%)
IV	11 (7.6%)
Pathologic type	
Serous adenocarcinoma	63 (43.8%)
Mucinous adenocarcinoma	35 (24.3%)
Endometrioid adenocarcinoma	28 (19.4%)
Clear cell carcinoma	18 (12.5%)

* Values are presented as mean ± standard deviation (range) or number (%). Abbreviations: HTN; hypertension, DM; diabetes mellitus, LN; lymph node.

**Table 2 cancers-17-00345-t002:** Association between grade of intravascular platelet aggregation in tumors and clinicopathologic characteristics of patients with ovarian carcinoma.

Value	Number of	Grade 1	Grade 2	Grade 3	*p*-Value
	Patients	(n = 25)	(n = 85)	(n = 34)	
Age (years)					
≤50	79	16	51	12	0.030
>50	65	9	34	12	
Tumor site					
Unilateral	71	12	44	15	0.745
Bilateral	73	13	41	19	
Ascites					
Absent	33	6	21	6	0.703
Present	111	19	64	28	
Pelvic LN metastasis					
Negative	105	19	64	22	0.467
Positive	39	6	21	12	
PALN metastasis					
Negative	99	17	60	22	0.819
Positive	45	8	25	12	
Pathologic type					
Serous	63	12	33	18	0.337
Mucinous	35	8	23	4	
Endometrioid	28	4	18	6	
Clear cell	18	1	11	6	
Stage					
I	51	12	35	4	0.002 *
II	14	6	5	3	
III	68	6	39	23	
IV	11	1	6	4	
Thrombocytosis ^+^					
Negative	135	25	78	32	0.325
Positive	9	0	7	2	
CTx response	80 ^#^				
Platinum-sensitive	66	10	38	18	0.451
Platinum-resistant	14	1	7	6	

* Spearman’s correlation analysis and all other analyses were performed with a chi-square test. ^+^ Thrombocytosis was defined as 450 × 10^3^/μL or higher. ^#^ From a total of 144 patients, 80 relapsed patients were found. Abbreviations: LN, lymph node; PALN, paraaortic lymph node; CTx, chemotherapy.

**Table 3 cancers-17-00345-t003:** Univariate analysis of 5-year and 10-year overall survival according to clinicopathologic characteristics and corresponding *p*-values (n = 144).

Value (Number of Patients)	5-Year OS	10-Year OS	*p*-Value
Age			
≤50 (n = 79)	68.4%	67.1%	<0.001
>50 (n = 65)	38.5%	33.8%	
Ascites			
Absent (n = 33)	81.8%	81.8%	0.001
Present (n = 111)	46.8%	43.2%	
Pelvic LN metastasis			
Negative (n = 105)	62.9%	60.0%	0.001
Positive (n = 39)	33.3%	30.8%	
Paraaortic LN metastasis			
Negative (n = 99)	62.2%	60.6%	0.001
Positive (n = 45)	33.3%	30.8%	
Stage			
I, II (n = 65)	86.2%	86.2%	<0.001
III, IV (n = 79)	29.1%	24.1%	
Thrombocytosis			
Negative (n = 135)	55.6%	52.6%	0.421
Positive (n = 9)	44.4%	44.4%	
Intravascular platelet aggregation in tumors			
Grade 1 (n = 25)	72.0%	72.0%	0.037
Grade 2 (n = 85)	55.3%	52.9%	
Grade 3 (n = 34)	41.2%	38.2%	

Abbreviations: OS, overall survival; LN, lymph node.

**Table 4 cancers-17-00345-t004:** Correlation between immunohistochemical staining grade for HIF-1α, VEGF, and PDGF and intravascular platelet aggregation in ovarian carcinoma (n = 144).

	HIF-1α					PDGF				VEGF			
	Grade	0	1	2	3	0	1	2	3	0	1	2	3
Intravascular	0	13	8	3	1	4	5	15	1	2	10	8	5
platelet aggregation	1	41	30	14	0	17	35	27	6	9	28	40	8
	2	10	8	13	3	4	11	14	5	1	8	18	7
Spearman’s correlation coefficient	0.226 *					0.034				0.131			
*p*-value	0.006					0.686				0.116			

The numbers in each cell of the table represent the number of patients corresponding to each level of intravascular platelet aggregation and expression of HIF-1α, PDGF, and VEGF. * The correlation coefficient between HIF-1α and intravascular platelet aggregation in ovarian carcinoma is 0.226, and the coefficient of determination (*r*^2^) is 0.051. Abbreviation: HIF, hypoxia-inducible factor; PDGF, platelet-derived growth factor; VEGF, vascular endothelial growth factor.

## Data Availability

Data are available from the corresponding authors upon reasonable requests.

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
