# Peer review of "Correlation Between Intravascular Platelet Aggregation in Tumors and Hypoxia-Inducible Factor 1 Alpha Expression in Epithelial Ovarian Cancer: Implications for Prognosis and Staging"

_cancers, 2025, doi:10.3390/cancers17030345_

Round 1
Reviewer 1 Report
Comments and Suggestions for Authors
Dear authors,
I read your study which aims to evaluate the association between tumor intravascular platelet aggregation and ovarian carcinoma prognosis and investigate underlying mechanisms, with interest. I have some comments:
1) The authors should include all the limitations of this study at the end of the discussion section
2) The end of the discussion is like a repetition of the conclusions reported in the conclusion section - please just merge the two texts in the conclusion section
3) I'm quite skeptical about the conduction of survival analysis for more reasons: 1) patients during follow-up could have incurred in different treatments and therapeutic pathways and prognosis of patients cannot depend entirely on platelet aggregation grade - had the authors conducted adjusted COX analysis by other variables or conducted stratified analysis (for example stage of the disease); 2) the data used by the authors are quite old (2000-2010) and survival could have changed deeply during the last 15 years since new therapeutic options came into the market; are these results generalizable also nowadays?
If these points are not well-motivated, I suggest authors remove survival analysis from the manuscript or deeply address the limitations in the limits section
Minor:
Line 274: Please also refer to the fact that the different prognosis can possibly depend also on the proangiogenic role of platelets in the tumor microenviroment (DOI: 10.3390/ijms232113401)
Author Response
Reviewer 1
Dear authors,
I read your study which aims to evaluate the association between tumor intravascular platelet aggregation and ovarian carcinoma prognosis and investigate underlying mechanisms, with interest. I have some comments:
à Thank you for taking the time to review our manuscript and for providing such detailed and constructive feedback. Your thoughtful comments and suggestions have been invaluable in helping us improve the quality and clarity of our study. We deeply appreciate your effort and dedication, which have allowed us to refine our work further.
1) The authors should include all the limitations of this study at the end of the discussion section
à We appreciate your suggestion to include the limitations of the study. There are several limitations in our study, but we did not describe them. Therefore, the sentence below was added to the bottom of the discussion. Please check it. (page 9, line 300-312)
“This study has several limitations. First, as a retrospective study relying on previously collected data, it may lack consistency or completeness. Second, the patients included in this study were treated between 2000 and 2010, a relatively long time ago, which may reduce the applicability of our findings to the current treatment landscape. While updated data from more recently treated patients could provide additional in-sights, the shorter follow-up periods associated with these patients may make it hard to perform a comprehensive survival analysis. Furthermore, we tried to minimize confounding factors introduced by relatively recent treatment modalities, such as poly (ADP‐ribose) polymerase inhibitors, which have significantly influenced patient out-comes. Finally, although our study results show a significant relationship between intravascular platelet aggregation and HIF-1α, the correlation is very weak. This suggests that in addition to the effects of hypoxic damage, other mechanisms may also have had an effect. Therefore, further studies are needed to understand the underlying mechanisms of this study result.”
2) The end of the discussion is like a repetition of the conclusions reported in the conclusion section - please just merge the two texts in the conclusion section
à I agree with your review comments. Duplicate parts have been deleted.
3) I'm quite skeptical about the conduction of survival analysis for more reasons:
3-1) patients during follow-up could have incurred in different treatments and therapeutic pathways and prognosis of patients cannot depend entirely on platelet aggregation grade - had the authors conducted adjusted COX analysis by other variables or conducted stratified analysis (for example stage of the disease);
à We understand your concerns regarding the survival analysis. To address these issues, we conducted new Kaplan-Meier survival analyses and performed univariate analysis of 5-year and 10-year overall survival based on clinicopathologic characteristics, calculating the corresponding p-values. Based on these results, multivariate analysis was then performed using the Cox regression hazard model. The results were added to Table 3 and the results section of the manuscript. Additionally, the original survival graph (originally figure 2) was removed, as it was deemed to potentially lead to an over-interpretation of the study results.
1) page 6, line 184-190
“Table 3 presents the results of the Kaplan-Meier survival analysis conducted based on CD42b IHC staining grades. Univariate analysis was performed to evaluate the association between the clinicopathologic characteristics with ovarian carcinoma patients and survival rates. As a result, age, ascites, pelvic or para-aortic lymph node metastasis, stage, and grade of intravascular platelet aggregation of the tumor were statistically significantly related to prognosis. The grade of intravascular platelet aggregation in ovarian carcinoma is statistically significantly associated with a poor prognosis (p=0.037).”
2) page 6, line 192-194
“Table 3. Univariate analysis of 5-year and 10-year overall survival according to clinicopathologic characteristics and corresponding p-values (n=144). à Please refer to the main text.”
3) page 7, line 196-205
“Based on the results of Table 3, multivariate analysis was performed for age (≤50 vs. >50 years), the presence of ascites and pelvic or para-aortic lymph node metastasis, stage (I, II vs. III, IV), and tumor intravascular platelet aggregation (grade 1 vs. grades 2 and 3) by using Cox regression hazard model. Age and stage were identified as independent prognostic factors for the survival of patients with ovarian carcinoma: age ≤50 vs. >50 (adjusted HR 1.940, 95% CI 1.197–3.146, p = 0.007) and stage I, II vs. III, IV (adjusted HR 6.874, 95% CI 3.369–14.025, p< 0.001). However, the presence of ascites, pelvic or para-aortic lymph node metastasis, and tumor intravascular platelet aggregation were not independent prognostic factors. This result was identical even when tumor intravascular platelet aggregation was categorized as grade 1, 2 vs. grade 3.”
3-2) the data used by the authors are quite old (2000-2010) and survival could have changed deeply during the last 15 years since new therapeutic options came into the market; are these results generalizable also nowadays? If these points are not well-motivated, I suggest authors remove survival analysis from the manuscript or deeply address the limitations in the limits section
à Thanks for the good point. As mentioned in the limitations, it was judged that if recent patients were enrolled, the follow-up period would be shortened and survival analysis would be difficult. In addition, factors such as PARPi, a relatively recently developed ovarian cancer treatment (estimated to have been adapted for treatment since around 2015), can have a significant impact on survival rates, so relatively old data were used to exclude these factors. These details have also been added to the limitations at the bottom of the discussion. (page 9, line 301-308)
“This study has several limitations. First, as a retrospective study relying on previously collected data, it may lack consistency or completeness. Second, the patients included in this study were treated between 2000 and 2010, a relatively long time ago, which may reduce the applicability of our findings to the current treatment landscape. While updated data from more recently treated patients could provide additional in-sights, the shorter follow-up periods associated with these patients may make it hard to perform a comprehensive survival analysis. Furthermore, we tried to minimize confounding factors introduced by relatively recent treatment modalities, such as poly (ADP‐ribose) polymerase inhibitors, which have significantly influenced patient out-comes. Finally, although our study results show a significant relationship between intravascular platelet aggregation and HIF-1α, the correlation is very weak. This suggests that in addition to the effects of hypoxic damage, other mechanisms may also have had an effect. Therefore, further studies are needed to understand the underlying mechanisms of this study result.”
Minor:
Line 274: Please also refer to the fact that the different prognosis can possibly depend also on the proangiogenic role of platelets in the tumor microenviroment (DOI: 10.3390/ijms232113401)
à Following your good review, I have revised the text and added the references. (page 9, line 264)
“According to the results in Table 3, tumor intravascular platelet aggregation is as-sociated with poor prognosis in patients with ovarian carcinoma. Additionally, based on the results in Table 2, tumor intravascular platelet aggregation was found to be as-sociated with advanced stage in ovarian carcinoma patients. This suggests that poor prognosis may also be associated to advanced stage of ovarian carcinoma. However, according to the results of multivariate analysis, intravascular platelet aggregation does not appear to be a direct survival factor in ovarian carcinoma. Based on these results, it is suggested that as the stage progresses, platelet aggregation within tumor blood vessels and subsequent exposure to hypoxia may impact prognosis. There are studies on association between tumor platelet infiltration and the prognosis of patients with pancreatic and gastric cancers [14-16]. One study has shown that platelet aggregation induced by tumor cells protects tumor cells from TNF-α-mediated cytotoxicity, which helps tumor cells evade the body's immune system [17]. However, the different prognosis may also possibly depend on the proangiogenic role of platelets in the tumor microenvironment [additional reference - Filippelli, A.; Del Gaudio, C.; Simonis, V.; Ciccone, V.; Spini, A.; Donnini, S. Scoping Review on Platelets and Tumor Angiogenesis: Do We Need More Evidence or Better Analysis? Int J Mol Sci 2022, 23, doi:10.3390/ijms232113401].”

Reviewer 2 Report
Comments and Suggestions for Authors
The manuscript investigates the correlation between ovarian carcinoma and the tumor intravascular platelet aggregation. Also, the authors hypothesize that tumor intravascular platelet aggregation induces microvascular thrombosis and hypoxia. To test these hypotheses the correlation between CD42b and HIF-1a, PDGF, VEGF was investigated.
The paper is well-written. However, the conclusion needs to be rethink in order to enforce the significance of the paper.
Major revision
Tumor cell-induced platelet aggregation helps cancer cells to evade the body’s immune system. The platelet aggregation protects tumor cells from TNF-α-mediated cytotoxicity (Philippe, Carole, et al. "Protection from tumor necrosis factor-mediated cytolysis by platelets." The American journal of pathology 143.6 (1993): 1713). Moreover, tumor–platelet aggregate embolizes the microvasculature at a new extravasation site (Malik AB. “Pulmonary microembolism.” Physiol Rev. 1983 Jul;63(3):1114-207. doi: 10.1152/physrev.1983.63.3.1114. PMID: 6348809), facilitates the adhesion of tumour cells to the vascular endothelium (Rickles, Frederick R., and Anna Falanga. "Molecular basis for the relationship between thrombosis and cancer." Thrombosis research 102.6 (2001): V215-V224) and releases a number of growth factors that can be used by tumor cells for proliferation (Honn KV, Tang DG, Chen YQ. “Platelets and cancer metastasis: more than an epiphenomenon.” Semin Thromb Hemost. 1992;18(4):392-415. doi: 10.1055/s-2007-1002578. PMID: 1470927). It would therefore be advisable to discuss some of these mechanisms in order to better explain the results obtained in the paper.
Minor revision
Table 4. Table 4 is difficult to understand. It is not clear which one is the grade related to immunohistochemistry staining for platelet aggregation and which one is the grades related to HIF-1a, PDGF, VEGF expression.
Author Response
The manuscript investigates the correlation between ovarian carcinoma and the tumor intravascular platelet aggregation. Also, the authors hypothesize that tumor intravascular platelet aggregation induces microvascular thrombosis and hypoxia. To test these hypotheses the correlation between CD42b and HIF-1a, PDGF, VEGF was investigated. The paper is well-written. However, the conclusion needs to be rethink in order to enforce the significance of the paper.
à Thank you for taking the time to review our manuscript and for providing constructive feedback. Your thoughtful comments and suggestions have been invaluable in helping us improve the quality and clarity of our study. We deeply appreciate your effort and dedication, which have allowed us to refine our work further. When we reviewed it again, we acknowledged that the conclusion was lacking. Accordingly, overlapping parts were deleted and modified.
Major revision
Tumor cell-induced platelet aggregation helps cancer cells to evade the body’s immune system. The platelet aggregation protects tumor cells from TNF-α-mediated cytotoxicity (Philippe, Carole, et al. "Protection from tumor necrosis factor-mediated cytolysis by platelets." The American journal of pathology 143.6 (1993): 1713).
Moreover, tumor–platelet aggregate embolizes the microvasculature at a new extravasation site (Malik AB. “Pulmonary microembolism.” Physiol Rev. 1983 Jul;63(3):1114-207. doi: 10.1152/physrev.1983.63.3.1114. PMID: 6348809), facilitates the adhesion of tumour cells to the vascular endothelium (Rickles, Frederick R., and Anna Falanga. "Molecular basis for the relationship between thrombosis and cancer." Thrombosis research 102.6 (2001): V215-V224) and releases a number of growth factors that can be used by tumor cells for proliferation (Honn KV, Tang DG, Chen YQ. “Platelets and cancer metastasis: more than an epiphenomenon.” Semin Thromb Hemost. 1992;18(4):392-415. doi: 10.1055/s-2007-1002578. PMID: 1470927). It would therefore be advisable to discuss some of these mechanisms in order to better explain the results obtained in the paper.
à The several references you provided were very helpful in estimating the mechanism according to the results of our paper. Accordingly, I have added a few references you suggested to the manuscript.
1)page 9, line 263
“One study has shown that platelet aggregation induced by tumor cells protects tumor cells from tumor necrosis factor-α mediated cytotoxicity, which helps tumor cells evade the body's immune system [additional reference - Philippe, Carole, et al. "Protection from tumor necrosis factor-mediated cytolysis by platelets." The American journal of pathology 143.6 (1993): 1713]”
2) page 9, line 274
“This increased risk is attributed to hypercoagulability caused by platelet aggregation [4,6, and additional reference - Rickles, Frederick R., and Anna Falanga. "Molecular basis for the relationship between thrombosis and cancer." Thrombosis research 102.6 (2001): V215-V224].”
3) page 9, line 277
“Cancer changes platelet behavior by triggering platelet granule release and modifying platelet phenotype and RNA profile. In particular, cancer cells induce platelet aggregation and this ability correlate to potential of cancer cell metastasis [6, and additional reference - Honn KV, Tang DG, Chen YQ. “Platelets and cancer metastasis: more than an epiphenomenon.” Semin Thromb Hemost. 1992;18(4):392-415. doi: 10.1055/s-2007-1002578.]”
Minor revision
Table 4. Table 4 is difficult to understand. It is not clear which one is the grade related to immunohistochemistry staining for platelet aggregation and which one is the grades related to HIF-1a, PDGF, VEGF expression.
à Table 4 has been revised to make it easier to understand. I have added Spearman's correlation coefficient and p-values at the bottom of the table to help with interpretation. Please take a check and confirm. (page 7, line 216 -224)
Reviewer 3 Report
Comments and Suggestions for Authors
This manuscript characterized the correlation between tumor intravascular platelet aggregation and the prognosis of ovarian cancer. The study was mainly based on the pathological and molecular analysis of samples from 144 ovarian cancer patients. Overall, the data analysis was clear; however, the manuscript did not clearly articulate the significance and impact of the findings - the observation is not strong enough to make it a therapeutic target, or a reliable biomarker. The authors should provide more data to support the hypothesis that intravascular platelet is not a result, but a driver of worse prognosis in patients and could have meaningful therapeutic implications.
Here are more detailed comments:
- Although Figure 3 shows the expression of HIF1alpha, PDGF, and VEGF are higher in tissue with higher grade of platelet aggregation levels, it does not align with the statistical results in Table 4, which showed only HIF1alpha has a significant correlation.
- Also, as the correlation between HIF1 alpha and platelet aggregation is only 0.226, it means the correlation is considered as weak (0.20-0.39). And this is the only one molecule that showed a significant correlation with platelet aggregation. The authors should provide stronger data to demonstrate the impact of platelet aggregation.
- In the discussion section, the manuscript (line 290) mentioned “Based on the correlation between HIF1alpha and tumor intravascular platelet aggregation in this study, it can be suggested that tumor intravascular platelet aggregation induces changes in the tumor microenvironment, promoting hypoxic damage within the tumor, which may contribute to reduced survival rates in cancer patients.” However, it is not shown in the manuscript that platelet aggregation is a driver that “induces” hypoxia condition or upregulation / overexpression of HIF1alpha. The data only showed a weak correlation. The authors need to conduct (at least) in vitro experiment to demonstrate this correlation
- In lines 239-246, the manuscript mentioned that “clonal thrombocytosis by myeloproliferative neoplasm such as essential thrombocythemia, chronic myelogenous lekemia, polycythemia vera, and primary myelofibrosis may also occur” and “occult malignancy was noted in nearly 40% of 140,000 patients with thrombocytosis but without inflammation or iron deficiency”. However, in table 3, only less than 10% samples / patients showed thrombocytosis. It’s hard to use the dysregulation in platelet levels in those studies to support the observation here. The author should not use a less directly associated pathological changes (thrombocytosis) to imply the “intravascular platelet aggregation” here
- In lines 293-295, the manuscript cited one paper (ref28) and mentioned that tumor intravascular platelet aggregation activated Hif1alpha and promoted drug resistance. However, ref 28 is a review paper and did not mention this finding explicitly. Please check the reference and provide a strong, direct, and reliable supporting study.
- In the entire study, only ovarian cancer tissue are analyzed. However, it is unclear if normal non-malignant tissue have similar levels of platelet aggregation. The analysis should include a small number of non-cancerous tissue as well.
Some minor comments:
- In the introduction section, the manuscript referred to studies that reported the risk of pulmonary thromboembolism is increased in OC, or platelet crosstalked to cancer cells to contribute to cancer progression. However, what is missing is the intravascular platelet aggregation - are there any previous studies? Why did the authors start with it vs. other platelet activities (e.g., thrombosis)
- In Figure 2, please also label the number of patients in each group in the legend (i.e., grade 1 n=25, grade 2 n=85) to help readers understand the size of each group without always referring back to the first table
- In Table 3 the r2 values should be provided
Author Response
This manuscript characterized the correlation between tumor intravascular platelet aggregation and the prognosis of ovarian cancer. The study was mainly based on the pathological and molecular analysis of samples from 144 ovarian cancer patients. Overall, the data analysis was clear; however, the manuscript did not clearly articulate the significance and impact of the findings - the observation is not strong enough to make it a therapeutic target, or a reliable biomarker. The authors should provide more data to support the hypothesis that intravascular platelet is not a result, but a driver of worse prognosis in patients and could have meaningful therapeutic implications.
à Thank you for taking the time to review our manuscript and for providing such detailed and constructive feedback. Your thoughtful comments and suggestions have been invaluable in helping us improve the quality and clarity of our study. We deeply appreciate your effort and dedication, which have allowed us to refine our work further. Following your review, we concluded that our study results might have been overinterpreted. Therefore, we have added and revised the discussion section as follows. (page 8-9, line 251-264)
“According to the results in Table 3, tumor intravascular platelet aggregation is as-sociated with poor prognosis in patients with ovarian carcinoma. Additionally, based on the results in Table 2, tumor intravascular platelet aggregation was found to be as-sociated with advanced stage in ovarian carcinoma patients. This suggests that poor prognosis may also be associated to advanced stage of ovarian carcinoma. However, according to the results of multivariate analysis, intravascular platelet aggregation does not appear to be a direct survival factor in ovarian carcinoma. Based on these results, it is suggested that as the stage progresses, platelet aggregation within tumor blood vessels and subsequent exposure to hypoxia may impact prognosis. There are studies on association between tumor platelet infiltration and the prognosis of patients with pancreatic and gastric cancers. One study has shown that platelet aggregation induced by tumor cells protects tumor cells from tumor necrosis factor-α mediated cytotoxicity, which helps tumor cells evade the body's immune system. However, the different prognosis may also possibly depend on the proangiogenic role of platelets in the tumor microenvironment.”
Here are more detailed comments:
- Although Figure 3 shows the expression of HIF1alpha, PDGF, and VEGF are higher in tissue with higher grade of platelet aggregation levels, it does not align with the statistical results in Table 4, which showed only HIF1alpha has a significant correlation.
à That figure is not a table showing the correlation between each factor (HIF, PDGF, VEGF) and platelet aggregation levels. This is just an example listing representative photos when reading the ratings for each element. As the existing figure 2 was deleted, that figure was modified to figure 2.
- Also, as the correlation between HIF1 alpha and platelet aggregation is only 0.226, it means the correlation is considered as weak (0.20-0.39). And this is the only one molecule that showed a significant correlation with platelet aggregation. The authors should provide stronger data to demonstrate the impact of platelet aggregation.
à I deeply agree with your opinion. I've added a limitation on this point at the bottom of the discussion. (page 9, line 308-312)
“This study has several limitations. First, as a retrospective study relying on previously collected data, it may lack consistency or completeness. Second, the patients included in this study were treated between 2000 and 2010, a relatively long time ago, which may reduce the applicability of our findings to the current treatment landscape. While updated data from more recently treated patients could provide additional in-sights, the shorter follow-up periods associated with these patients may make it hard to perform a comprehensive survival analysis. Furthermore, we tried to minimize confounding factors introduced by relatively recent treatment modalities, such as poly (ADP‐ribose) polymerase inhibitors, which have significantly influenced patient out-comes. Finally, although our study results show a significant relationship between intravascular platelet aggregation and HIF-1α, the correlation is very weak. This suggests that in addition to the effects of hypoxic damage, other mechanisms may also have had an effect. Therefore, further studies are needed to understand the underlying mechanisms of this study result.”
- In the discussion section, the manuscript (line 290) mentioned “Based on the correlation between HIF1alpha and tumor intravascular platelet aggregation in this study, it can be suggested that tumor intravascular platelet aggregation induces changes in the tumor microenvironment, promoting hypoxic damage within the tumor, which may contribute to reduced survival rates in cancer patients.” However, it is not shown in the manuscript that platelet aggregation is a driver that “induces” hypoxia condition or upregulation / overexpression of HIF1alpha. The data only showed a weak correlation. The authors need to conduct (at least) in vitro experiment to demonstrate this correlation
à We admit that we overinterpreted our findings. However, there were several previous in vitro studies that tumor intravascular platelet aggregation induces changes in the tumor microenvironment, promoting hypoxic damage within the tumor. And there was another previous in vitro study that platelet aggregation activates HIF-1α. Our study results suggest that, clinically, tumor intravascular platelet aggregation is associated with patient survival rates (Table 3). These findings imply a potential link with the advanced stage of ovarian carcinoma (Table 2) and a possible association with hypoxic damage although weak correlation. I hope you understand this.
- In lines 239-246, the manuscript mentioned that “clonal thrombocytosis by myeloproliferative neoplasm such as essential thrombocythemia, chronic myelogenous lekemia, polycythemia vera, and primary myelofibrosis may also occur” and “occult malignancy was noted in nearly 40% of 140,000 patients with thrombocytosis but without inflammation or iron deficiency”. However, in table 3, only less than 10% samples / patients showed thrombocytosis. It’s hard to use the dysregulation in platelet levels in those studies to support the observation here. The author should not use a less directly associated pathological changes (thrombocytosis) to imply the “intravascular platelet aggregation” here
à Thank you for your good comment. Our findings study intravascular platelets within ovarian carcinoma cancer cells. Our study does not suggest an association between thrombocytosis on CBC (Complete blood cell) and tumor intravascular platelets. We include this sentence in manuscript because include information about the relationship between malignancy and platelet. However, because we acknowledge that our manuscript sentences may be occur misunderstanding, we have modified the first paragraph of the discussion. (page 8, line 236-242)
“The association between cancer and platelets has been consistently reported. Thrombocytosis often arises as a reactive condition triggered by factors such as acute infection, tissue damage, chronic inflammation, surgery, iron deficiency, or the rebound effect after bone marrow suppression. Although thrombocytosis is not specific to cancer, the association between platelets and cancer has been increasingly recognized. In patients without anemia or inflammation, thrombocytosis may indicate the presence of occult malignancies.”
- In lines 293-295, the manuscript cited one paper (ref28) and mentioned that tumor intravascular platelet aggregation activated Hif1alpha and promoted drug resistance. However, ref 28 is a review paper and did not mention this finding explicitly. Please check the reference and provide a strong, direct, and reliable supporting study.
à It seems that our reference was not enough to contain reference content. The text and references have been modified as follows. Instead of a review paper, an experimental paper was presented. Please check. (page 9, line 291-294)
“Tumor intravascular platelet aggregation has the potential to activate HIF-1α, which contributes to enhancing drug resistance by promoting cellular metabolic changes and inhibiting cell death [additional reference - Li JQ, Wu X, Gan L, Yang XL, Miao ZH. Hypoxia induces universal but differential drug resistance and impairs anticancer mechanisms of 5-fluorouracil in hepatoma cells. Acta Pharmacol Sin. 2017 Dec;38(12):1642-1654. doi: 10.1038/aps.2017.79. Epub 2017 Jul 10. PMID: 28713155”
- In the entire study, only ovarian cancer tissue are analyzed. However, it is unclear if normal non-malignant tissue have similar levels of platelet aggregation. The analysis should include a small number of non-cancerous tissue as well.
à Thank you for your insightful comment. However, our study is not designed to compare platelet aggregation levels between normal ovarian tissue and ovarian cancer tissue. Instead, our research specifically focuses on investigating the correlation between intravascular platelet aggregation within ovarian cancer tissue and patient prognosis. We aimed to explore how platelet aggregation within the tumor microenvironment influences survival outcomes, rather than analyzing differences between malignant and non-malignant tissues. I hope you understand this.
Some minor comments:
- In the introduction section, the manuscript referred to studies that reported the risk of pulmonary thromboembolism is increased in OC, or platelet crosstalked to cancer cells to contribute to cancer progression. However, what is missing is the intravascular platelet aggregation - are there any previous studies? Why did the authors start with it vs. other platelet activities (e.g., thrombosis)
à We acknowledge the lack of explanation regarding the background of our study. We have added references and more contents about the background of our research to the manuscript. Please confirm. (page 2, line 50-52)
”It is known that cancer cells can induce platelet aggregation, which is caused by crucial molecules or surface receptor molecules on the surface of cancer cells [6]. According to previous study, platelet aggregation in tumor cells is associated with cancer progression and metastasis, leading to research on new platelet-targeted therapeutic strategies [7]. Based on this background, we hypothesized that tumor intravascular platelet aggregation may induce occlusion of tumor micro vessels and hypoxia of the cancer microenvironment in patients with ovarian carcinoma. As a result, cancer cells are subjected to hypoxic stress, which is associated with proliferation, metastasis, and chemoresistance, which may be associated with a poor prognosis.”
- In Figure 2, please also label the number of patients in each group in the legend (i.e., grade 1 n=25, grade 2 n=85) to help readers understand the size of each group without always referring back to the first table
à Figure 2 (survival analysis according to the grade of CD42b IHC staining) has been removed due to concerns about overinterpreting our research results. Instead, we newly presented the univariate analysis of 5-year and 10-year overall survival according to clinicopathologic characteristics and the corresponding p-values in Table 3. Please check.
- In Table 3 the r2 values should be provided
à I think you are referring to table4. Am I right? If correct, coefficient of determination (r2) has been added to the bottom of the table. Please confirm. (page 7, line 222)

Round 2
Reviewer 1 Report
Comments and Suggestions for Authors
Thank you for reply to all my comments. The manuscript is now improved
Reviewer 2 Report
Comments and Suggestions for Authors
Dear authors,
The revision of the manuscript was appropriate and fullfill my requests.